# Wind Imagery in *Shijing*: Sacrificing to the Wind God in Early China

**Chao Cai** [1,2,*] and **Siu Kwai Yeung** [2]

1 School of Foreign Studies, Shaoguan University, Shaoguan 512005, China
2 Faculty of Education, University of Macau, Macau 999078, China
* Correspondence: mgccily@163.com

**Abstract:** *Shijing* 詩經 is the earliest collection of Chinese poems and songs traditionally considered to be compiled by Confucius. Scholarship on this collection deems the widespread wind imagery contained in it to be either a metaphor for males or a medium of emotional expression. However, the religious ideas involved in the sacrifices to the wind gods in early China, which are, in fact, deeply linked with the "wind" in *Shijing*, warrant further consideration. This article focuses on the relation between the "wind" in *Shijing* and the religious ideas involved in sacrificial rites ("*ningfeng* 寧風" and "*difeng* 禘風") to the wind gods. Drawing upon the history of wind disasters and sacrifices to the wind gods in early China, this article suggests that the pieces entitled "*Gufeng* 谷風" (included in the *Xiaoya* 小雅 section) and "*Herensi* 何人斯" provide descriptions of "*ningfeng*" (appeasing unwanted wind). Moreover, it argues that the pieces entitled "*Kaifeng* 凱風" and "*Tuoxi* 蘀兮" depict a genial wind in connection with harvest, childbearing, and prosperity involved in "*difeng*".

**Keywords:** *Shijing*; wind god; ningfeng; difeng; religion in early China; sacrifice

## 1. Introduction

The wind is a widespread natural phenomenon closely related to human survival. It brings nourishing rain to crops or stirs up storms that cause destruction. However, the ancient people could not understand the causes of wind as a natural phenomenon. They believed in the existence of wind gods who had direct control over the wind. As such, these gods had to be pleased or placated through worship rituals. In the oracle-bone inscriptions (OBI), there are many records that mention the rituals around (e.g., divinations) and sacrifices to the wind gods (Li 2020). For instance:

> Di sacrifices will be conducted to the four quarters: The (wind god of) eastern quarter is called Xi (divider), the wind (it blows) is called Xie (harmonious); the (wind god of) southern quarter is called Yin (gentle), the wind (it blows) is called Wei (calm); the (wind god of) western quarter is called Wei (afflitor), the wind (it blows) is called Yi (confronting); the (wind god of) northern is called Fu (lurker), the wind (it blows) is called Yi (pestilence). 帝( 禘) 于東方曰析, 風曰協; 帝于南方曰因, 風曰夷; 帝于西方曰彝, 風曰韋; 帝于北方曰伏, 風曰役. (Li 1989, pp. 25–26)[1]

This inscription reveals that in early China, the wind was conceived of as being controlled by the wind gods in the four quarters (in relation to the four directions). Scholars agree that the "four quarters" do not merely indicate the directions of the wind, but also match the four seasons (Hu 1956; Li 1985; Feng 1994). The wind could be either *xiefeng* 協風, which brings warmth in spring, the cold north wind in winter, or even a hurricane or storm that causes destruction. Because of the worship of and feelings of fear toward the wind gods, a large number of rituals and sacrifices were conducted to venerate or propitiate the wind gods. This led to the emergence of wind imagery in the earliest collection of Chinese poems and songs, namely *Shijing* 詩經.[2] As David Schaberg has pointed out,

"that song outlives mortals, carrying news of their [ancient people's] glories and sufferings to future generations, is an idea found everywhere in early literature" (Schaberg 1999, p. 305).

## 2. The Worship of the Wind Gods and the Wind Imagery in *Shijing*: A Review of Relevant Scholarship

Modern scholars have delved into the potential relationship between *Shijing* and ancient myths. Wen Yiduo 聞一多 (1899–1946) has argued that the "wind" in *Shijing* is a metaphor for males, which is derived directly from the worship of the wind gods in early China (Wen 1993a). In early inscriptions, the "winds" and "wind gods" are expressed by the same single character, which has been transcribed as "*feng* 鳳" (phoenix) in the field of OBI studies. The wind god "looks like a bird with a 辛-like crown and elaborate tassel-like wings" (Li and Takashima 2022, p. 86) and acts as an agent carrying nourishing rain and seasonal change. This is tantamount to linking the wind to the fertility deity, thus implying sexuality and becoming a metaphor for males.

In contrast, Shirakawa Shizuka 白川靜 (1910–2006) contends that the "wind" in *Shijing* is present in connection with "the wind gods in the four quarters 四方風神" and stands as a symbol of feelings and emotions:

> In early China, the wind was conceived as being controlled by the wind gods in different quarters; thus, different wind gods (in different quarters) blow different kinds of wind. 風は古代においては、それぞれの方位によって、方位神に駆使される風神であり、それ自身それぞれの性格を持つものであった. (Shirakawa 1999, p. 433)

> The wind, along with other natural phenomena, influences human emotions; thus becomes a medium of emotional declaration. 風などの天象が人間の感情に種々の影響を与え、従って感情の表現にそれらが用いられることは極めて自然である. (Shirakawa 1999, p. 431)

According to Shirakawa, the fierce wind in "*Zhongfeng* 終風" (a song included in the *Beifeng* 邶風 section) is a way of alluding to anxiety at the beginning of the song. The fierce wind then becomes a whirlwind of dust, which symbolizes the traumas of a broken marriage experienced by a woman. Meanwhile, in "*Gufeng* 谷風", a song included in the *Xiaoya* 小雅 (Lesser Court Hymns) section, the strong wind from the valley is also associated with the bitterness of a wife's feelings (Shirakawa 1999, p. 432). However, Ye Shuxian 葉舒憲 (1954-) disagrees and instead proposes a different interpretation:

> Shirakawa Shizuka defines the wind in "*Zhongfeng*" by anxiety and explains "*zhongfeng qie bao, guwo ze xiao* 終風且暴, 顧我則笑" (the wind blows and is fierce, he looks at me and smiles)[3] as a description of a man who behaved rudely and was about to abandon his wife. This is a big mistake. 白川靜說《終風》中的"風"是"不安的暗示"; 又說"終風且暴, 顧我則笑"是對動作粗暴, 準備遺棄女方的男子的哀嘆, 大誤. (Ye 1994, p. 594)

> The song "*Zhongfeng*" illustrates an ideal male from a woman's perspective using the natural elements of wind, haze, clouds, and thunder. 《終風》一首則兼風, 霾, 雲, 雷四種意象形容一個女性愛情幻想中的配偶形象. (Ye 1994, p. 593)

Following Ye Shuxian's interpretation, the wind, along with the notion of the wind as a birth catalyst in ancient mythology, is brought into line with the symbolism of primitive sexuality. In other words, Ye Shuxian interprets the "wind" in the same manner as Wen Yiduo, connecting its imagery in *Shijing* with the archetypal symbol of the wind gods.

Other researchers have also noted the wind imagery in *Shijing*. For instance, Rong Xiaocuo and Liu Yuan agree that the "wind" in *Shijing* is rooted in the worship of the wind gods, and they further associate the wind imagery with the themes of love and marriage (Rong 2002; Liu 2019), maternal love (Rong 2002), and social suffering (Liu 2019).

Previous studies have pointed out that the wind imagery in *Shijing* is undoubtedly drawn from the rich fountain of the ancient worship of the wind gods revealed by the OBI. "Sacrificing to the wind gods is often encountered in OBI" (Li and Takashima 2022, p. 81),

which consists of two types of sacrificial rites: "*difeng* 禘風" and "*ningfeng* 寧風". "*Difeng*" refers to large, regularly scheduled sacrifices to the wind gods conducted by the emperor (Li 2020), with the purpose of *qinian* 祈年 (Hu 1956; Rao 1988; Li 2020). The reading of *qinian* 祈年 as "praying for (a year of) a bumper harvest 祈豐年" is clear in Zheng Xuan's 鄭玄 (127–200) commentary (Mao and Zheng 2018, p. 426). In contrast, "*ningfeng*" refers to propitiating the wind gods to appease unwanted wind (Li and Takashima 2022). However, the question of whether the wind imagery in *Shijing* is associated with the religious ideas involved in "*difeng*" and "*ningfeng*", as well as with the history of wind disasters in early China, has not yet received enough attention. There are resonances between the representations of the wind in *Shijing* and OBI that can provide important clues to answer this question.

### 3. The Wind Imagery in *Shijing*

*Shijing* 詩經, the earliest anthology of Chinese poetry, contains 305 poems and songs dating from the Western Zhou (1046 B.C.–771 B.C.) to the Spring and Autumn period (770 B.C.–476 B.C.). In the *Shijing*, there are seventeen songs in which the "wind" plays a symbolic and descriptive role. As shown in Table 1, the "wind" is rendered as genial or gentle only in the songs entitled "*Kaifeng* 凱風", "*Tuoxi* 蘀兮", and "*Zhengmin* 烝民". In contrast, there is an abundance of negative imagery pertaining to the wind, which is described as "fierce wind", "violent wind", "rushing wind", "cold wind", and even "wind and rain". For instance, in the pieces entitled "*Zhongfeng* 終風", "*Bifeng* 匪風", "*Herensi* 何人斯", "*Siyue* 四月", and "*Sangrou* 桑柔", the wind is depicted as carrying strong or even threatening power and is closely associated with the poets' anguish. Remarkably, there are two songs entitled "*Gufeng* 谷風", one of which is included in the *Beifeng* 邶風 (Odes of Bei) section and the other in the *Xiaoya* 小雅 (Lesser Court Hymns) section. Both songs begin with "*xixi gufeng* 習習谷風", which describes a gust coming from a great valley with harsh imagery. However, the *Maozhuan* 毛傳, a seminal work of annotation and commentary of *Shijing* completed in the Han Dynasty (202 B.C.-220), interpreted "*xixi gufeng* 習習谷風" as "the soft wind of spring 習習, 和舒貌, 東風謂之谷風" (Mao and Zheng 2018, pp. 50, 293). James Legge (1815–1897) follows the interpretation of "*xixi gufeng*" offered in the *Maozhuan*:

> Gently blows the east wind,\With cloudy skies and with rain.\[Husband and wife] should strive to be of the same mind,\And not let angry feelings arise 習習谷風, 以陰以雨。黽勉同心, 不宜有怒. (Legge 1960, p. 55)

According to Legge's translation, "*xixi* 習習" describes the gentle breath of the wind; thus, "*gufeng* 谷風" is interpreted as "the east wind that brings clouds and rains, and all genial influences" (Legge 1960, p. 55). From this perspective, the first two lines (*xixi gufeng, yiyin yiyu* 習習谷風, 以陰以雨) are the poet's allusions to the harmony and happiness of the family. However, Yan Can 嚴粲 (1197–?) explains the lines very differently, as referring to "the angry demonstrations of the husband, like the gusts of wind coming constantly from great valleys and bringing darkness and rain ( 故風, 雅二谷風) 言以陰以雨, 喻暴怒" (Yan 1983, pp. 2–3). Yan Can's interpretations of "*xixi*" and "*gufeng*" have been widely accepted by modern scholars who propose to study the poems and songs independently from the political and historical interpretations that the Han scholars gave them:

> "*Xixi*" is the sound of strong wind. However, in the *Maozhuan* 毛傳 and *Zhengjian* 鄭箋, "*xixi*" is explained as genial and soft, while "*gufeng*" is defined as the east wind, which are just opposite to their original meanings. 習習本大風之聲, 《傳》、箋》以谷風為東風, 訓習習為和舒、和調, 揆之《詩》意, 皆適得其反. (Wen 1993a, p. 371)

> "*Xixi*", similar to "*sasa* 颯颯" (a reduplicative conveying the sounds of the wind), describes the sound of constant strong wind, while "*gufeng*" refers to the gusts of wind coming from the valleys. 習習, 猶颯颯, 連續不斷的大風聲; 谷風, 來自山谷的大風. (Cheng and Jiang 1991, pp. 91, 624)

The English translation by Arthur Waley (1889–1966), an outstanding scholar famous for translating Chinese and Japanese poetry, also supports Yan Can's reading:

Zip, zip the valley wind,\Bringing darkness, bringing rain.\Strive to be of one mind,\Let there be no anger between you. 習習谷風, 以陰以雨。黽勉同心, 不宜有怒. ([Waley 1937](), p. 100)

Compared to Legge's translation, Waley's translation of "*gufeng*" is more consistent with the theme of the sorrow of a wife abandoned by her husband and supplanted by another woman ([Cheng and Jiang 1991](), p. 90; [Zhu 2011](), p. 28). Therefore, the "wind" in "*Gufeng* 谷風" should be interpreted as gusts of wind coming from valleys, because it is more suitable for expressing the sorrow of a wife and her appealing to the gods for mercy, which can be exemplified by *Shuowen jiezi* 説文解字. According to *Shuowen jiezi* 説文解字, an ancient Chinese dictionary complied by Xu Shen 許慎 (58-147 CE), the phoenix (wind god) "resides in caves and empty valleys 暮宿風穴" ([Xu 1981](), p. 148). Xu Shen has further pointed out that "caves and empty valleys are where the wind comes out 風穴, 風所從出也" ([Xu 1981](), p. 148). A perfect example in *Shijing* linking the wind gods with empty valleys is a hymnic song entitled "*Sangrou* 桑柔" included in the *Daya* 大雅 (Major Court Hymns) section:

Great winds have a path,\They come from the large empty valleys 大風有隧, 有空大谷. ([Legge 1960](), p. 525)

In this sense, we can conclude that the gust of wind coming from valleys (*gufeng* 谷風) is metaphorical or allusive, referring to both the situational context (of an abandoned wife) and the ancient worship of the wind gods.

On the other hand, in the piece entitled "*Beifeng* 北風", the wind is rendered as the cold north wind:

Cold blows the north wind. 北風其涼. ([Legge 1960](), p. 67)

The cold north wind in "*Beifeng*" symbolizes "a tyrannical government" ([Cheng and Jiang 1991](), p. 113) and implies anguish resulting from social upheavals. According to Zheng Xuan 鄭玄, "cold wind is a hazard to everything 寒涼之風, 病害萬物" ([Mao and Zheng 2018](), p. 60). The cold wind in the song "*Lüyi* 綠衣" is linked with the husband who suffers anguish at the death of his wife. Moreover, the song "*Fengyu* 風雨" highlights a painful separation after marriage through the use of wind and rain imagery. In all the aforementioned pieces, the "wind" is deeply linked with distress, whether social or personal. Meanwhile, in another song, "*Chixiao* 鴟鴞", the "wind" is present as thrashing and rocking the dwelling places and appears to offer more negative and threatening imagery closely related to emotional suffering and even survival.

In fact, most of the wind imagery in *Shijing* is tied to its strong power or its relation to other natural elements, such as severe cold and rain, and is closely linked with anguish and sorrow. In consideration of the abundance of negative imagery pertaining to the "wind" in *Shijing*, this article outlines the history of wind disasters and the sacrifices to the wind gods in early China as an important context for understanding the religious ideas embodied in the wind imagery in *Shijing*.

**Table 1.** Summary of the symbolic and descriptive role of the wind discussed in Section 3. (English translation is cited from James Legge, *The Chinese Classic: The She King; or, the Book of Poetry*. Hong Kong: Hong Kong University Press, 1960).

| Title of the Song | The Lines (or Stanza) in Which the Wind Plays a Symbolic and Descriptive Role |
| --- | --- |
| *Lüyi* 綠衣, | 絺兮綌兮, 淒其以風。Linen, fine or coarse,\Is cold when worn in the wind (p. 42). |
| *Zhongfeng* 終風 | 終風且暴, 顧我則笑。The wind blows and is fierce,\He looks at me and smiles (p. 46). |
| *Kaifeng* 凱風 | 凱風自南, 吹彼棘心。The genial wind from the south,\Blows on the heart of that jujube tree (p. 50). |
| *Gufeng* 谷風 (included in the *Beifeng* 邶風 section) | 習習谷風, 以陰以雨。Gently blows the east wind,\With cloudy skies and with rain (p. 55). |
| *Beifeng* 北風 | 北風其涼, 雨雪其雱。Cold blows the north wind,\Thick falls the snow (p. 67). |
| *Tuoxi* 蘀兮 | 蘀兮蘀兮, 風吹其女。Ye withered leaves! Ye withered leaves!\How the wind is blowing you away (p. 138)! |
| *Fengyu* 風雨 | 風雨淒淒, 雞鳴喈喈。Cold are the wind and the rain,\And shrilly crows the cock (p. 143). |
| *Bifeng* 匪風 | 匪風發兮, 匪車偈兮。顧瞻周道, 中心怛兮。Not for the violence of the wind,\Not for the rushing motion of a chariot.\But when I look to the road to Chow,\Am I pained to the core of my heart (p. 218). |
| *Chixiao* 鴟鴞 | 予室翹翹, 風雨所漂搖。My house is in a perilous condition,\It is tossed about in the wind and rain (p. 235). |
| *Sijian* 斯干 | 椓之橐橐, 風雨攸除。T'oh-t'oh went on the pounding,\Impervious [the walls] to the wind and rain (p. 304). |
| *Herensi* 何人斯 | 彼何人斯, 其為飄風。What man was it?\He is like a violent wind (p. 345). |
| *Gufeng* 谷風 (included in the *Xiaoya* 小雅 section) | 習習谷風, 維山崔嵬。無草不死, 無木不萎。Gently blows the east wind,\And on the rock-covered tops of the hills,\There is no grass which is not dying,\No tree which is not withering (p. 350). |
| *Liao'e* 蓼莪 | 南山烈烈, 飄風發發。民莫不穀, 我獨何害？Cold and bleak is the southern hill,\The rushing wind is very fierce.\People all are happy,\Why am I alone thus miserable (p. 352)? |
| *Siyue* 四月 | 冬日烈烈, 飄風發發。民莫不穀, 我獨何害？The winter days are very fierce,\And the storm blows in rapid gusts.\People all are happy,\Why do I alone suffer misery (p. 357)? |
| *Quane* 卷阿 | 有卷者阿, 飄風自南。Into the recesses of the large mound,\Came the wind whirling from the south (p. 491). |
| *Sangrou* 桑柔 | 大風有隧, 有空大谷。Great winds have a path,\They come from the large empty valleys (p. 525). |
| *Zhengmin* 烝民 | 吉甫作誦, 穆如清風。仲山甫永懷, 以慰其心。I, Yin Keih-hoo, have made this song,\May it enter like a quiet wind.\Among the constant anxieties of Chung Shan-foo,\To soothe his mind (p. 545). |

## 4. The History of Wind Disasters and Sacrifices in Early China

The earliest historical records of wind disasters go back to the Yin and Shang Dynasties (see Figure 1).

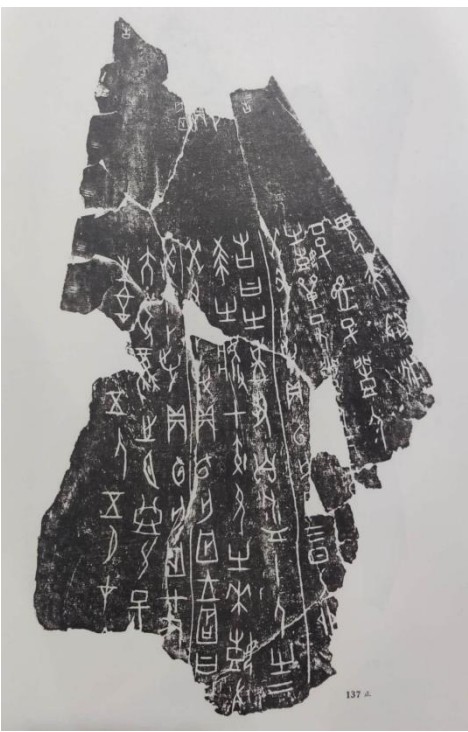

**Figure 1.** An oracle-bone inscription in which a strong rushing wind in the Shang dynasty is documented (Guo and Hu 1982, p. 32).

Above is a bone inscription made from a bovine's shoulder blade that records a strong rushing wind in the late Shang dynasty: "On the day of *jiachen* 甲辰, a strong rushing wind swept 甲辰 … （有）大驟風( 鳳)" (Zhu 2020).[4] Such a record provides important information about wind disasters in early China, from which ancient people's feelings of fear can be seen. Apart from the OBI, records of wind disasters can also be found in ancient Chinese texts. For instance, the "*Jinteng* 金滕" chapter of *Shangshu* 尚書 (*Book of Documents*) describes a wind disaster as follows:

> In autumn, the crops were ripe and had not yet been harvested. Suddenly came the lightning and fierce wind; the grain fell, and the trees were uprooted; people were terrified. 秋, 大熟, 未獲, 天大雷電以風, 禾盡偃, 大木斯拔, 邦人大恐. (*Shangshu Zhengyi* 2007, p. 501)

Moreover, *Liji* 禮記 (*Book of Rites*), a collection of descriptions of social norms, administration, and ritual matters during the Zhou dynasty (1046 B.C.-256 B.C.), also mentions wind disasters:

> The implementation of winter decrees in mid-fall results in frequent wind disasters. （仲秋）行冬令, 則風災數起. (*Liji* 1995, p. 123)

Additionally, *Guanzi* 管子, a political and philosophical text in the Warring States period (476 B.C.-221 B.C.), documents the wind and rain and their impact on agriculture in the "*Xiaokuang* 小匡" chapter:

> The violent wind and rain swept; thus, neither the harvest of the crops nor the livestock could flourish. 飄風暴雨數臻, 五穀不蕃, 六畜不育. (*Guanzi Jiaozhu* 2004, p. 426)

Early historical literature, such as the *Shangshu* and *Guanzi*, provides valuable insights into the history of wind disasters. As stated in the chapter of "*Zhouyu* 周語" (one chapter of *Guoyu* 國語), "agriculture plays a fundamental role for the well-being of common people of *Zhou* 夫民之大事在農" (Zuo 2020, p. 44).[5] Because crops are totally dependent on the weather, ancient people were very fearful of catastrophic weather events. As a result, they developed an elaborate cult of the wind gods to give them a measure of predictability and

protection. This is why wind imagery is so widespread in *Shijing*. The strong winds in *Shijing* are illustrated through reduplicative words such as "*xixi* 習習" and "*fafa* 發發". These words, on the one hand, serve primarily as the sounds of the strong winds; on the other hand, they are used against the danger of overlooking and forgetting the strong winds, deploying "the mnemonic function of language" (Owen 2001, p. 287).

Undoubtedly, the sacrificial rites held to pray to the wind gods are inextricably linked with the feelings of fear of strong winds. *Zhouli* 周禮 (*Rites of the Zhou*) documents the name of the wind god and the details of the sacrifices to the wind god as follows:

> Make the burnt offering to *sizhong*, *siming*, *fengshi*, and *yushi*. 以榔燎祀司中, 司命, 飆師, 雨師. (*Zhouli Jinzhu Jinyi* 1972, p. 192)

"*Feng* 飆" is a variant form of "*feng* 風" (wind); thus, "*fengshi* 飆師" refers to "*fengshi* 風師", the name of the wind god in early China. Meanwhile, "*youliao* 榔燎" refers to the burnt offerings.[6] On the other hand, *Zhouli* 周禮 also mentions "*ningfeng* 寧風" as follows:

> *Xiaozhu* (minor invocator) is in charge of small sacrificial ceremonies and the prayer words for *hou* 侯, *rang* 禳, *dao* 禱, and *si* 祀. The purpose is to pray for good luck, a good harvest, and good rain, as well as for the strong wind and drought to subside. 小祝掌小祭祀將事侯禳禱祠之祝號, 以祈福祥, 順豐年, 逆時雨, 寧風旱. (*Zhouli Jinzhu Jinyi* 1972, p. 266)

*Zhouli* 周禮 reveals the various officials involved in the ritual and sacrificial system in early China. In accordance with *Zhouli*, *xiaozhu* 小祝 (minor invocator) assists "*dazhu* 大祝" (great invocator) and is in charge of the prayer words through which communication with the nature gods and other powers is thought to take place, including the prayer words to wind gods to request that the strong wind subside. According to the *Erya* 爾雅 (also known as *Erh-ya*, the first surviving Chinese dictionary), "sacrificing to the wind gods was named *zhe* ( 磔) 祭風曰磔" (Guo and Xing 2000, p. 200). Guo Pu 郭璞 (276–324) further pointed out, "*zhe* 磔 is similar to the custom of *Jin* 晉 to offer a dog as a sacrificial animal to appease the unwanted wind 今俗當大道中磔狗, 雲以止風, 此其像" (Guo and Xing 2000, p. 200).

All these historical records show that sacrificing to the wind gods was prevalent in early China. As discussed above, there are two types of sacrificial rites to the wind gods in the ritual and sacrificial system in early China: "*ningfeng* 寧風" and "*difeng* 禘風". Therefore, the religious ideas involved in "*ningfeng*" and "*difeng*" should be taken into consideration as we look deeper into the wind imagery in *Shijing*.

## 5. The Relation between the Wind Imagery in *Shijing* and the Religious Ideas Involved in "*Ningfeng*" and "*Difeng*"

### 5.1. Ningfeng 寧風

As already mentioned, "*ningfeng*" refers to unscheduled sacrificial activities in the hope of appeasing strong wind. According to Yu Xingwu 于省吾 (1896–1984), there were a number of sacrificial rites pertaining to "*ning* 寧" (appease) in early China:

> The oracle-bone inscription documents ancient sacrifices such as "*ningfeng*", "*ningyu*", "*ningshui*", "*ningji*", and "*ningzhong*". The "*feng* (wind)", "*yu* (rain)", "*shui* (water)", "*ji* (disease)", and "*zhong* (locust)" were undesired kinds that caused damage. Thus, sacrifices were held to pray to the gods to wish they would subside. 卜辭有寧風, 寧雨, 寧水, 寧疾, 寧蟲之祭. 謂風, 雨, 水, 疾, 蟲為害, 祈禳於神祇以求其止息. (Yu 1999, p. 2662)

It is apparent that "*ningfeng*" is one of "*ningji* 寧祭" (appeasing sacrifice). Martin Kern has argued that many hymns from *Shijing* "not only epitomize the ideal of ritual order but also on occasion provide elaborate descriptions of ritual acts" (Kern 2005, pp. x–xi). As shown in Table 1, the piece entitled "*Gufeng*" (included in the *Xiaoya*— 小雅 Lesser Court Hymns—section) suggests that the wind, depicted as gusts coming from the valleys, is in close relation to the religious ideas involved in "*ningfeng*". The gusts of wind, "followed by rain and thunder 維風及雨, 維風及穨"[7], cause disasters that "no grass but is dying, no tree but is wilting 無草不死, 無木不萎". The prayers to the wind god are expressed

through "*jiangkong jiangju* 將恐將懼" (in the time of fear and dread) and "*jiangan jiangle* 將安將樂" (in your time of rest and pleasure) over and over again, aiming at propitiating the wind god.

Compared with "*Gufeng*", "*Herensi* 何人斯" illustrates more concrete aspects of "*ningfeng*". "*Herensi*", also included in the *Xiaoya*, is arranged in eight stanzas of six lines each. First, in stanzas three and four, we read:

> What man was it?\Why came he to the path inside my gate?\I heard his voice,\But did not see his person.\He is not ashamed before men,\He does not stand in awe of Heaven 彼何人斯, 胡逝我陳, 我聞其聲, 不見其身, 不愧於人, 不畏於天

> What man was it?\He is like a violent wind.\Why came he not from the north?\Or why not from the south?\Why did he approach my dam,\Doing nothing but perturb my mind 彼何人斯, 其為飄風, 胡不自北, 胡不自南, 胡逝我梁, 祇攪我心. ([Legge 1960](), pp. 344–45)

Stanzas three and four both begin with "*bi heren si* 彼何人斯", which is followed by "*wowen qisheng, bujian qishen* 我聞其聲, 不見其身" and "*qi wei piaofeng* 其爲飄風". The former distinguishes the wind god from humans, as he appears only through his voice (sound); and the latter fragment tells us that the ritual performer was attracting the attention of a god who is causing "*piaofeng* 飄風" (a violent, threatening wind). Then, in stanza seven:

> Here are the three creatures [for sacrifice],\And I will take an oath to you 出此三物, 以詛爾斯. ([Legge 1960](), p. 346)

According to *Maozhuan*, "three creatures 三物" refers to three types of sacrificial animals: porcine ( 豕), canine ( 犬), and fowl ( 鷄) ([Mao and Zheng 2018](), p. 288). Some researchers have pointed out that "*ningfeng*" is accompanied by sacrificial offerings, mainly animals ([Li and Takashima 2022]()). A good example can be found in OBI, "*guihaibu: yunan ningfeng tunyi* 癸亥卜: 于南寧風豕一" (literally, "cracked on *guihai*: an appeasing sacrifice to the wind god will be conducted toward the south with the sacrificial offering of one porcine") ([Yu 1999](), p. 2659). The sacrifice of other types of animals is attested to by another OBI that reads: "*jiaxu zhen: qi ningfeng sanyang sanquan santun* 甲戌貞: 其寧風三羊三犬三豕" (literally, "divined on *jiaxu*: an appeasing sacrifice to the wind god will be conducted with the use of three sheep, three dogs, and three pigs") ([Yu 1999](), p. 2659). The above examples suggest that the sacrificial offering of animals is one of the requirements for conducting "*ningfeng*". Therefore, we can conclude that the text of "*Herensi*" embodies authentic actions of the appeasing sacrifice "*ningfeng*" and the religious notion of dissuading the wind god from causing destructive wind.

The provenance of "*Herensi*" remains disputed. According to the *Xiaoxu* 小序 (*Minor Prefaces*), "*Herensi*" is written by a duke of *Su* 蘇, who had been slandered by a duke of *Bao* 暴 ([Mao and Zheng 2018](), p. 286). However, Zhu Xi 朱熹 and other scholars doubt this historical interpretation given in the *Minor Prefaces*, saying it is an over-interpretation because there is no evidence that "*Herensi*" concerns the duke of *Su* ([Fang 1986](), p. 413; [Zhu 2011](), p. 189). Some scholars have proposed different interpretations; Cheng Junying 程俊英 (1901–1993) and Jiang Jianyuan 蔣見元 (1950–), for instance, interpret "*Herensi*" as a song elaborating the rupture between two friends.

Despite a lack of unified interpretation of "*Herensi*" ([Pan 2004]()), the text provides a comprehensive account of the appeasing sacrifice to the wind god from an outside perspective, relating the religious ideas of "*ningfeng*" to the poet's own situational context. It can, therefore, be understood as a descriptive account of "*ningfeng*".

*5.2. Difeng* 禘風

As mentioned above, the purpose of "*difeng* 禘風" is "*qinian* 祈年" (praying for a rich harvest). According to *Erya* 爾雅, "*di* 禘" refers to "large sacrifices (to the nature gods and ancestral spirits) 禘, 大祭也" ([Guo and Xing 2000](), p. 201). Thus, "*difeng*" refers to large sacrificial rites to the wind gods in the hope of a bumper harvest. In connection

with the purpose of "*difeng*", scholars further developed the idea that "*difeng*" explains the prosperity of all living things, because the wind gods in four quarters, to whom the sacrificial rites of "*difeng*" were conducted, are closely related to the four seasons (Hu 1956; Ye 1994). Moreover, the religious ideas involved in "*difeng*" are not merely associated with harvest, but also imply childbearing and prosperity. Even though no songs in *Shijing* could be identified as either a descriptive account of "*difeng*" or a text to be performed in sacrificial rites of "*difeng*", there are two songs, "*Kaifeng* 凱風" and "*Tuoxi* 蘀兮", in which the wind imagery is closely associated with the religious ideas involved in "*difeng*". In stanzas one and two of "*Kaifeng*":

> The genial wind from the south\Blows on the heart of that jujube tree \Till that heart looks tender and beautiful\What toil and pain did our mother endure 凱風自南, 吹彼棘心, 棘心夭夭, 母氏劬勞

> The genial wind from the south\Blows on the branches of that jujube tree\Our mother is wise and good\But among us there is none good 凱風自南, 吹彼棘薪, 母氏聖善, 我無令人. (p. 50)

"*Kaifeng*" is a name given to the south wind in *Maozhuan* 毛傳 and Zhu Xi's 朱熹 (1130–1200) influential *Shi ji zhuan* 詩集傳. "*Kaifeng*", with its genial influences on all vegetation, is the "triumphant or pleasant wind" (Legge 1960, p. 50), serving as a metaphor for a mother who struggles to raise her children. "*Jixin* 棘心" refers to the tender shoots of the jujube tree and symbolizes young children. James Legge interprets "*jixin*" as "the heart of that jujube tree", because the heart of the tree is the most inner and hidden part and is "more difficult for the genial influence to reach" (Legge 1960, p. 50). Even so, the south wind nurtures the tender shoots to grow into branches. *Erya* 爾雅 also supports this reading: "the south wind nourishes all living things, thus is called *Kaifeng* 南風長養, 萬物喜樂, 故曰凱風" (Guo and Xing 2000, p. 192). Thus, it is plausible to conclude that, in the song "*Kaifeng*", the wind imagery is rooted in the religious ideas involved in "*difeng*". Another example illuminating the relation between "*Kaifeng*" and "*difeng*" is "*you zi qiren* 有子七人" (we are seven sons), a line from stanza three, which is closely related to the prayer words for an abundance of progeny. In other words, the wind imagery in "*Kaifeng*" is present in an obvious relationship to childbearing and prosperity.

In another song entitled "*Tuoxi* 蘀兮" (arranged in two stanzas of four lines each), the wind imagery is also related to "*difeng*":

> Ye withered leaves! Ye withered leaves\How the wind is blowing you away\O ye uncles\Give us the first note, and we will join in with you 蘀兮蘀兮, 風其吹女, 叔兮伯兮, 倡予和女

> Ye withered leaves! Ye withered leaves\How the wind is carrying you away\O ye uncles\Give us the first note, and we will complete [the song] 蘀兮蘀兮, 風其漂女, 叔兮伯兮, 倡予要女. (Legge 1960, p. 138)

The *Minor Preface* gives a historical interpretation of "*Tuoxi*". Many scholars, however, have pointed out that the song "*Tuoxi*" does not concern historical figures or events (Zhu 2011; Cheng and Jiang 1991). "*Tuoxi*" is a courtship song sung by young women and men. According to Legge, "*tuo* 蘀" refers to a tree with its leaves withered and ready to fall (Legge 1960, p. 138). However, Yuan Mei 袁梅 (1924–2017) has argued that "*tuo* 蘀" refers to a plant as a tender reed 蘀為草名, 指初生蘆葦 and further pointed out that "*Tuoxi*" can be viewed as a song performed by both young women and men in a spring ritual (Yuan 1985, pp. 257–58). The "wind" in "*Tuoxi*" stands as a symbol of men, and "*tuo* 蘀" acts as a metaphor for women (Cheng and Jiang 1991, p. 242). The "wind" here is a catalyst for vegetation sprouting and is further associated with childbearing and prosperity. Furthermore, some researchers have set out to reconstruct the religious rite embodied in the song "*Tuoxi*". For instance, Wu Quanlan 吳全蘭 (1968–) pointed out that "*Tuoxi*" contains actual speeches (also songs) uttered by both young women and men who participated in a religious ritual (Wu 2003). As already mentioned, in early China, a large number of religious rituals were conducted to please the nature gods to act in favor of the ritual performers (e.g., rulers or

ministers) and their states. Meanwhile, marriage was also an important theme in the religious rituals during that period, especially in spring (Zhu 2017), a season that was often described in relation to marriage in early historical literature. For instance, the "*Diguan* 地官" chapter of *Zhouli* 周禮 (*Rites of the Zhou*) reads: "spring is a good time for marriage, and elopement is also allowed 中春之月, 令會男女, 於是時也, 奔者不禁" (*Zhouli Jinzhu Jinyi* 1972, p. 144). Marriage customs in ancient Chinese culture underwent many changes. In early China, people were encouraged to marry freely in the spring to increase the population, as population growth is closely linked to the political and economic development of an ancient society (Zhu 2017). In this sense, we can conclude that the "wind" in the courtship song "*Tuoxi*" becomes personified and is referred to repeatedly in the context of romance, marriage, and childbearing, in which the archetype of the wind is uniquely tied to the religious ideas of "*difeng*".

Additionally, each stanza in "*Tuoxi*" begins with the verse "*tuo xi tuo xi* 蘀兮蘀兮". This "A/B/A/B" pattern is an expressive repetition that appeared in the *Guofeng* 國風 (Airs of the States), *Xiaoya* 小雅 (Lesser Court Hymns), *Song* 頌 (Eulogia) sections (Yuan 2019). Even though this repetitive pattern is not confined to the *Ya* 雅 and *Song* 頌 sections, it is employed much more frequently in religious hymns than in other songs (Xia 1998).[8] Although it is very difficult to trace the exact meaning of the song "*Tuoxi*", it can be inferred that the wind imagery in "*Tuoxi*" is deeply rooted in the religious ideas and the context of the sacrificial rites of "*difeng*".

## 6. Conclusions

Although the wind imagery in *Shijing* has attracted some scholarly attention, limited research has been conducted to explore the relation between this wind imagery and the ancient sacrifices to the wind gods in early China. In our discussion, we found it necessary to investigate the history of wind disasters in early China before proceeding to further analysis. As Fa Li and Ken-ichi Takashima remind us, the ways in which ancient Chinese people coped with the threatening winds offers a window into their religious ideas (Li and Takashima 2022, p. 95). These religious ideas provided an effective backdrop for approaching the wind imagery in *Shijing*, since poetry served as an essential means of preserving ideas and knowledge in early literature (Schaberg 1999).

By bringing the descriptions of wind disasters and sacrifices in early China and the wind imagery together, we can gain a glimpse into the religious ideas embodied in the wind imagery in *Shijing*. Compared with the genial and quiet wind, the "wind" with strong or even threatening power appears with a much higher frequency in *Shijing*, which is consistent with the abundant descriptions of wind disasters in early China.

As discussed in the present study, the hymnic song entitled "*Herensi*" can be identified as a descriptive account of "*ningfeng*", as it provides elaborate descriptions of appeasing violent wind, thus explaining the origin of this hymn. Moreover, "*Herensi*" should not be seen as the only text linked with "*ningfeng*". The imagery of the wind coming from valleys also appears to be closely associated with the religious ideas involved in "*ningfeng*". As for "*difeng*", the wind in the songs entitled "*Kaifeng*" and "*Tuoxi*" is depicted as genial as well as quiet, images that squarely meet the connotations of harvest, childbearing, and prosperity in "*difeng*".

**Author Contributions:** Conceptualization, C.C. and S.K.Y.; Investigation, C.C.; Resources, C.C.; Writing—original draft, C.C.; Writing—review & editing, S.K.Y. All authors have read and agreed to the published version of the manuscript.

**Funding:** This research received no external funding.

**Institutional Review Board Statement:** Not applicable.

**Informed Consent Statement:** Not applicable.

**Data Availability Statement:** Not applicable.

**Conflicts of Interest:** The authors declare no conflict of interest.

## Notes

[1]  The interpretations of the inscriptions cited in the paper are from *Selected Interpretations of Oracle-Bone Inscriptions* 甲骨文選注 and *Jiagu Wenzi Gulin* 甲骨文字詁林 [*Interpretation of Oracle-Bone Characters*]. In both sources, the inscriptions have been transcribed into modern Chinese characters in the most generally practiced way in the Oracle-Bone Inscription studies. See (Li 1989; Yu 1999).

[2]  *Shijing* 詩經, also known as *Shih-ching*, is one of the Five Classics 五經 and has been translated in the West variously as the Book of Poetry, Book of Songs, or Book of Odes.

[3]  Cited from the English translation by James Legge (1815–1897). See (Legge 1960, p. 46). The English translation of the songs cited in this paper mainly comes from James Legge.

[4]  The full text about the strong rushing wind is as follows: Crack making on *guimao* 癸卯 day, Zheng divined: There will be no disasters in the coming 10-day week. On *jiachen* 甲辰 day, a strong rushing wind swept; in that evening of *jiachen* 甲辰 day (followed by *Yisi* 乙巳 day), five people lost. In the fifth month; (it was) at Dun 敦 (that we carried out this divination). 癸卯卜, 爭貞：旬亡憂。甲辰 ... 大驟風( 鳳), 之夕皿( 向) 乙巳 ... 逸 ... 五人。五月, 才( 在)[ 敦]. See (Guo and Hu 1982, p. 32).

[5]  *Guoyu* 國語 is an ancient text that comprises 240 speeches dating from about 976 B.C. to about 453 B.C. While the author of *Guoyu* is unknown, some scholars have argued that Zuo Qiuming 左丘明 (556-451 B.C. or 502-422 B.C.) compiled it. The "*Zhouyu* 周語" (*Discourses of Zhou*), covering the Zhou court, is the first part of *Guoyu*.

[6]  "*Youliao* 槱燎" refers to burnt offerings, for example, making the burnt offering to the gods with the use of one bovine.

[7]  It has been argued by Wen Yiduo that "Tui 頹" refers to thunder (" 維風及頹", 猶言維風及雷也). See (Wen 1993b, p. 419).

[8]  "*Ya* 雅" is a joint name for the *Xiaoya* 小雅 (Lesser Court Hymns) and *Daya* 大雅 (Major Court Hymns) sections. Kern has pointed out that the vast majority of the banquet songs and religious hymns are included in *Ya* 雅 (Court Hymns) and *Song* 頌 (Eulogies) sections. See (Kern 2000, p. 49).

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
