# Peer review of "Wind Imagery in Shijing: Sacrificing to the Wind God in Early China"

_religions, doi:10.3390/rel14010102_

Round 1
Reviewer 1 Report
I found this to be a well-written and convincing article, effectively and efficiently presenting an argument for considering the religious implications of wind references in several shijing poems. There seems to be a thorough engagement with scholarship from the modern period while taking also into consideration several of the most influential classical interpretations. Overall, I found the article’s suggestion that the practice of ningfeng should be taken into account when interpreting “Herensi” quite interesting and compelling, and the same goes for the notion of difeng in the context of “Tuoxi.” If the article were developed further, I think a new interpretation of the poems could be offered given the new religious context being suggested for understanding the poems, but that seems to be beyond the scope of the task at hand.
As a side note, I also found the heavy inclusion of Chinese absolutely crucial in helping the reader understand the critical discussion of the poems’ content; it enhances the clarity of the arguments, which would be much more opaque if only Pinyin and English translations were used.
There are just a few typos I noticed, and a few wordings I think could be changed:
Line 6: no comma after “collections”
Line 9: change “have not been given enough” to “warrant further”
Line 23: delete the word “that”
Line 134: capitalize the C in the name “Yan Can”
Line 320: delete the word “indeed”
Author Response
We appreciate the reviewer’s positive comments. More importantly, your comments and suggestions have helped us improve the quality of our manuscript. We included a detailed response of how we attended to address the comments and suggestions. Please find the attached file “Response to reviewer 1 comments”.

Reviewer 2 Report
See the attached.

Author Response
We appreciate the reviewer’s insightful comments. More importantly, your comments are rigorous and insightful, enabling us to improve the quality of our manuscript. We included a detailed response of how we attended to address your comments and suggestions. Please find the attached file “Response to reviewer 2 comments”.

Reviewer 3 Report
This is a fascinating and important topic. As the author shows, the "wind" theme occurs frequently in the Shijing. And the author is well aware of other external evidence that can help illuminate the religious significance of the theme and its role in various poems. I'm impressed by the author's general knowledge and erudition.
Unfortunately, though, the paper is not really written in English! Consider the last sentence:
As for “difeng”, the pieces entitled “Kaifeng” and “Tuoxi” demonstrate a genial and quiet wind in relation to the harvest, as well as the connotations of pregnancy, fertility, and prosperity involved in “difeng”.
If I go back and look at the earlier discussion I can figure out what this means and could translate it into English, but the author has not done that.
Moreover, even if the English were clearer, it is not really clear what the thesis is. Being worried about wind-related disasters does nto really count as a "Religious" idea. A couple of poems discuss a "gentle wind"? Ok....But for an academic article you need to try to think about what that means: what is the broader significance of this, either historical, or cultural, or literary?
One way to do this would be to look more closely at some of the Western scholarship cited. The author cites Kern and Schaberg but does not really seem to have paid attention to what they are saying. You can't just cite them as background; you need to try to present a thesis that makes sense in relation to the arguments other scholars are making.
Another thing totally ignored in this paper is that wind (also criticism, 諷) is totally central to the theory of the Shijing. So there is much more to say about this, and one really needs to cite some scholarship on the 大序 and on the theory of early poetry as well.
So some good basic ideas, but much more work needs to be done for this to be a solid academic article in English.
Author Response
We appreciate the reviewer’s insightful comments. More importantly, your comments are rigorous and insightful, enabling us to improve the quality of our manuscript. We included a point-by-point response to your comments. Please find the attached file “Response to reviewer 3 comments”.

Round 2
Reviewer 2 Report
The major issues have been addressed in the revised manuscript. The paper looks nice in its present form now.
Author Response
We sincerely appreciate the reviewer’s positive comments on our revised manuscript. Your comments were very helpful and enabled us to improve the quality of our manuscript. The newly revised version has been carefully revised by a native English speaker to avoid causal language and to improve grammar and readability. Again thanks very much for your insightful comments and feedback.
Reviewer 3 Report
To be published in English, the paper needs to be rewritten completely. I just cited one example sentence before but the point is that it needs to be rewritten to make a coherent argument in English. Rather than just putting difficult terms in pinyin, they have to be explained and contextualized and translated.
I'm only taking the time to point this out again because I do think the content of the paper is great and well worth publishing. But you can't just do a literal translation of your Chinese article and publish it in English. You need to rewrite it to suit English-language academic expectations.
Author Response
We appreciate the reviewer’s positive comments and constructive feedback on our manuscript. We included a point-by-point response to your comments. Please find the attached file “Response to reviewer 3 comments”.
